# Modern Biodegradable Plastics—Processing and Properties: Part I

**DOI:** 10.3390/ma13081986

**Published:** 2020-04-24

**Authors:** Janusz Sikora, Łukasz Majewski, Andrzej Puszka

**Affiliations:** 1Department of Technology and Polymer Processing, Faculty of Mechanical Engineering, Lublin University of Technology, Nadbystrzycka 36, 20-618 Lublin, Poland; l.majewski@pollub.pl; 2Department of Polymer Chemistry, Institute of Chemical Sciences, Faculty of Chemistry, Maria Curie-Sklodowska University in Lublin, ul. Gliniana 33, 20-614 Lublin, Poland; andrzej.puszka@umcs.pl

**Keywords:** polymer processing, starch, extrusion, biodegradability, PLA, TPS, DSC, MFR

## Abstract

This paper presents a characterization of a plastic extrusion process and the selected properties of three biodegradable plastic types, in comparison with LDPE (low-density polyethylene). The four plastics include: LDPE, commercial name Malen E FABS 23-D022; potato starch based plastic (TPS-P), BIOPLAST GF 106/02; corn starch based plastic (TPS-C), BioComp^®^BF 01HP; and a polylactic acid (polylactide) plastic (PLA), BioComp^®^BF 7210. Plastic films with determined geometric parameters (thickness of the foil layer and width of the flattened foil sleeve) were produced from these materials (at individually defined processing temperatures), using blown film extrusion, by applying different extrusion screw speeds. The produced plastic films were tested to determine the geometrical features, MFR (melt flow rate), blow-up ratio, draw down ratio, mass flow rate, and exit velocity. The tests were complemented by thermogravimetry, differential scanning calorimetry, and chemical structure analysis. It was found that the biodegradable films were extruded at higher rate and mass flow rate than LDPE; the lowest thermal stability was ascertained for the film samples extruded from TPS-C and TPS-P, and that all tested biodegradable plastics contained polyethylene.

## 1. Introduction

The rapid increase in demand for inexpensive materials has caused an increased consumption and production of polymers to hundreds of millions of tons per annum. A consequence of this trend is a heavy pollution of the environment by the accumulation of polymer waste therein [1,2]. The largest contributor to this problem is the packaging industry, which primarily uses cheap petroleum-derived polyolefins *en masse* to produce disposable packaging film. The packaging industry is currently dominated by applications of many different polyethylene types due to the low costs of the material, a wide range of processing parameters, good mechanical properties, and relatively good thermal and chemical resistance [2,3]. However, natural degradation of polyethylene in natural environment progresses very slowly, similar to the case of other petroleum-derived polymers. It is assumed that complete degradation of polyethylene exposed to the natural environment might take several hundred years [2,4]. During the process, the material is gradually defragmented in size by abiotic and biotic phenomena, up to a point at which the polyethylene particle size becomes potentially hazardous to humans [5]. This process of defragmentation can also release toxic chemicals or trace elements from the colours applied used in the original plastic products [2,6]. Landfilling of polymer waste, which now a predominant method of disposal, is highly ineffective. Incineration of polymer waste requires high capital expenditure for a suitable processing infrastructure and results in emissions of harmful gaseous chemicals to the atmosphere [3,7,8].

The problem of high environmental pollution with plastics was acknowledged over 20 years ago and has been followed by attempts to reduce the use of plastics through recycling [7,8,9]. However, each recycling and reprocessing run of polymers reduces their properties due to thermal degradation, increasingly restricting the applications of the recovered material. It is then not recommended to recycle and reprocess virgin polymers more than three times [2,7,8]. Not all polymer plastics can be recycled. Hence, there has been a growing interest in biodegradable and biocompostable plastics derived from renewable material sources, which do not pollute the environment by virtue of partial or complete biotic degradation [10,11]. The development and production of biodegradable plastic materials seem to be the least expensive and most effective ways to efficiently manage plastic waste. To make biodegradable/compostable (in accordance with the definitions of biodegradability and compostability, presented in the standards EN 14046 and EN 13432) plastics, is a feasible full substitute for traditional petroleum-derived plastics (or at least reduce the consumption of the latter), they need a competitive edge based on performance properties or production output. The major hindrance to mass application of biopolymers is often a complex and stringent process of sourcing and processing, resulting in end product prices much higher than for petroleum-derived plastics [12].

The biopolymers currently applied in the industry can be classified in several ways; the main classification is based on the method of sourcing. Hence, there are synthetic biopolymers or biomass-derived biopolymers [10,13,14].

Widely used example of biopolymers synthesized from renewable material sources is polylactide (PLA) [10]. PLA is a polyester which can be derived with a number of methods, resulting in different molecular masses and material properties. Short-chain polymers are produced by condensation polymerisation of lactic acid; longer polymer chains, which provide better strength properties, are synthesized by polymerization with monomeric lactide ring opening [10,15,16]. Given its very good mechanical properties, rigidity, optical properties, biocompatibility, biodegradability, thermal resistance and extremely good processability, PLA has a high potential for industrial application. In addition, numerous physical and chemical modifications are currently used to improve PLA processability, so it can be successfully processed with high efficiency by injection moulding, heat forming and extrusion, including production of plastic film sheets by pour casting and blown film extrusion [10,17]. However, plastic film made form unmodified PLA sees limited packaging applications; it is very brittle and relatively poor in resistance to permeation by gases, especially oxygen [18]. It is why plastic film is often made from blends with a PLA matrix [19,20,21]. References state that a small addition of exfoliated mineral fillers can improve flexibility in PLA [22,23], whereas PLA mixed with polyhydroxybutyrate (PHB) effectively reduces permeability to gases, while improving resistance to moisture, making the PLA film, produced with this formulation, an attractive packaging material [24,25].

Biomass-derived biopolymers, which are currently applied on a large scale, are primarily polysaccharides and proteins of plant or animal origin. Examples, include cellulose, casein, chitin, and starch [10,26,27,28]. Starch has found most versatile use in plastic processing, although its application in the natural form is infeasible [29,30]. Native starch, a direct product from natural sources, has higher glass transition and melting temperatures than its thermal decomposition temperature; this property requires pre-treatment of native starch for downstream processing [31,32]. Pre-treatment of native starch into TPS (thermoplastic starch) is a complex process with multiple preconditions, in order to obtain a material with specified properties. These preconditions include: An optimum procedure of native starch drying [33], the quantities and types of plasticizers [34], an optimum technology for processing TPS and optimized processing conditions [29,35], and an optimum selection of the structural parts for the extrusion machine’s plasticizing system [36,37] to ensure a satisfactory homogenization level. Reference literature provides many examples of plasticizers with a proven effectiveness in application with native starch and which perform in compliance with internal or external plasticizing mechanisms. The examples of plasticizers include water, glycerine, urea, polyethylene glycol, citric acid, and sorbitol [10,38,39,40]. The types and quantity of applied plasticizers depend on the processing method and the type and origin of native starch. Native starch is a polysaccharide composed of linearly structured amylose and branched amylopectin [41]. The quantity ratio of both components is important to the degree of crystallinity and the crystalline grid type, the grain size, and the supermolecular structure [42]. For example, the share of amylose is 20–25% in potato starch, 25–30% in corn starch, and 17% in tapioca starch. Consequently, these differences are translated into different performance of thermoplastic starch, depending on its native starch origin [43,44,45]. Despite its many benefits, TPS is seldom used as a target material for direct production of finished goods, especially thin-walled ones, like plastic film. The main reasons, include poor flowing properties, a narrow range of processing parameters, a tendency for absorbing water, poor strength parameters, and high degradability of starch exposed to high temperatures, and its intense shearing in the plasticizing systems of extrusion machines [46,47]. Because of this, polymer blends with starch are used for production of packaging materials. Starch is used with other biopolymers (e.g., PLA or PCPA) [48,49] or petroleum-derived plastics (LDPE or LLDPE); in the latter case, a compatibilizer (a.k.a. coupling agent) is required due to differences in affinity to water between both components [50,51,52,53]. Blends of TPS and PE are successfully applied for blown film extrusion, where the addition of PE greatly improves processability and strength of the resulting plastic film [54,55]. The material is not fully biodegradable, although reference literature state that TPS/LDPE blends with already 60% of starch as fit for composting [53].

The aim of the extensive research, conducted by the authors of this work, was to provide a process-wise and utility-wise comparison of plastic films produced from a traditional petroleum-based plastic, or LDPE, and from biodegradable, compostable polymers based on starch derived from renewable material sources, with a determination of feasibility of each polymer material considered here for applications in the packaging industry. As biopolymers were used: potato starch based plastic (TPS-P), BIOPLAST GF 106/02; corn starch based plastic (TPS-C), BioComp^®^BF 01HP; and a polylactic acid (polylactide) plastic (PLA), BioComp^®^BF 7210, whereas the films were obtained by blowing extrusion. This work focuses on determining thermal properties (thermogravimetric and differential scanning calorimetry), chemical structure (Fourier Transform Infrared Spectroscopy (FTIR)) of the granulates and obtained films, as well as the rheological properties of the granulates. In addition, the geometrical properties of the obtained films and parameters describing the blown film extrusion process (depending on the rotation speed of the extruder screw) were determined. Based on the results obtained, the possibilities of processing and using the biodegradable materials, tested in the packaging industry, were determined. In the next work (being a continuation of these studies), the mechanical, optical, barrier and microscopic properties of the obtained films will be characterized.

## 2. Experimental

### 2.1. Materials

The blown film extrusion was performed with four different materials: LDPE, commercial name Malen E FABS 23-D022; potato starch based plastic (TPS-P), BIOPLAST GF 106/02; corn starch based plastic (TPS-C), BioComp^®^BF 01HP; and a polylactic acid (polylactide) plastic (PLA), BioComp^®^BF 7210.

The Malen E FABS 23-D022 LDPE was manufactured by Lyondell Basell (Rotterdam, The Netherlands) and intended for production of packaging film with a sheet thickness above 25 µm. With an anti-blocking agent and a lubricant, the film can be produced for application in automatic packaging machines. LDPE can be processed by casting or blown film extrusion. The manufacturer recommends processing temperatures between 160 °C and 220 °C. MALEN E FABS 23-D022 is primarily applied in production of plastic bags and upright liners, shrink-wrap film, film sleeves, and food packaging film [56].

BIOPLAST GF 106/02 manufactured by BioTec (Emmerich am Rhein, Germany) with TPS-P is fully biodegradable and fit for composting. It is free from plasticizers; hence, there is no input material deposition or emission of vapour in processing. This TPS-P plastic can be processed by heat forming, injection moulding, slot extrusion, and blown film extrusion. The manufacturer recommends processing with traditional LDPE plasticizing screws and within a temperature range of 140–180 °C. BIOPLAST GF 106/02 is primarily applied in production of disposable or short-lived goods, including food packaging materials, plastic film for agriculture, shopping bags or waste bin liners. Plastic film manufactured from BIOPLAST GF 106/02 is fully biodegradable and resists oil, grease, petrol and water; it is a good printing substrate for flexography and offset printing processes [57].

BioComp^®^BF 01HP belongs to a family of bioplastics manufactured by MicroTec (Pianiga, Italy) from biodegradable components of organic origin. This TPS-C plastic is characterised by optimum levels of moisture and plasticizers for successful melting and good flowability during processing. BioComp^®^BF 01HP is a novel plastic material designed primarily for blown film extrusion. The TPS-C plastic is suitable for processing in all types of blown film extrusion or pour casting lines, with standard operating settings of the extruders. Processing the material with traditional LDPE plasticizing screws, and within a temperature range of 150–170 °C, is recommended. The plastic comprises lactic acid (in the form of PLA) and corn starch. The primary application is production of shopping bags. The manufacturer guarantees the product is biodegradable at least in 90% over 6 months. To improve the processing properties of this plastic material, anti-blocking and lubricant agents made from biodegradable polyesters and lactic acid can be used [58].

Similar to the bioplastic specified above, BioComp^®^BF 7210 by MicroTec is primarily intended for processing by blown film extrusion. This PLA plastic is suitable for processing in all traditional blown film extrusion or casting lines with standard operating settings of plasticizing screws and extruders. The plastic comprises lactic acid (PLA) with talc. It is starch-free. A specially developed formulation provides the material with transparency and makes it a good choice for production of shopping bags. The recommended processing temperature range of BioComp BF 7210 is 140–180 °C. The processability of this material is closely related its moisture content; it is recommended to store it dedicated, sealed bags and consume the material in 6 h after unsealing [59]. Table 1 lists selected properties of the processed plastic materials.

### 2.2. Test Stand

The plastic film was extruded with a MB 45/750 single-screw extruder provided with a standard, traditional screw for LDPE plasticizing and the following specifications: screw diameter D = 45 mm, screw service length L = 28D, manufacturer: Kween B LTD., (Taipei, Taiwan). The extruder had 4 heating zones: at the extrusion head, at the plastic filter, and two in the plasticizing system, both complete with air fans for temperature stability control. The screw drive system allowed control over the screw rotational speed. The extruder was used with a spiral core cross-head and an extrusion nozzle with O.D. 60 mm and 1 mm in width for blown film extrusion, in which the film forms a sleeve blown freely upward. The blown film extrusion line featured a flattener and a film sleeve clamp, followed by a film windup unit with a stepless speed control of the windup rollers.

### 2.3. Research Programme and Methodology

In the researched blown film extrusion process, the plasticizing temperatures for each of the plastic material tested were individually selected on the basis of the manufacturer’s data and unified across all heating zones of the extruder. The plasticizing temperature values were thus as follows: LDPE—160 °C, TPS-P—160 °C, TPS-C—155 °C, and PLA—150 °C.

A variable of the blown film extrusion process was the rotational speed of the extruder screw, whereas all other processing parameters which defined the geometric features of extrusion blown film were set to produce film with a layflat film width, within 35–36 cm, and a single-ply thickness, within 20–25 µm. The interval of sampling test strips from each film was 15 s. With the foregoing assumptions, the relationships between the selected blown film extrusion process characteristics were developed as a function of the screw rotational speed, and the effect of film processing on the specific material’s characteristics of the film were determined.

The experimental research included:
Testing of thermal properties by application of differential scanning calorimetry (DSC) and thermogravimetric analysis. The DSC was performed with a DSC 204 ***F1***
*Phoenix^^®^^* machine manufactured by Netzsch (Günzbung, Germany) and Netzsch Proteus test data post-processing software (Version 6.0.0), where the heating and cooling rates were assumed at 10 °C/min within a temperature range of 150–180 °C (or 200 °C for the maximum limit). Each DSC sample was processed with two heating cycles. All DSC tests were made in an aluminium crucible with a pierced lid and in an argon gas shield (supplied at a rate of 20 mL/min). The reference container was an empty aluminium crucible. The *T*_g_ values of the test specimens were determined at the temperature of thermal curve inflection point. The melting point of the test specimens (*T*_m_) was determined as the maximum of the endothermic peak. The values of melting enthalpy (Δ*H_m_*) and degree of crystallinity (*X*_%_) were determined. The degree of crystallinity was calculated using the following equation,
(1)$$X_{\%}=\frac{\Delta H_{m}}{\Delta H_{m}^{0}}\times 100\%$$
where Δ*H_m_* is the melting enthalpy and $\Delta H_{m}^{0}$ is the melting enthalpy of 100% crystalline PE (293 J/g) [36], and PLA (93 J/g) [60], respectively. In the TPS test specimens the degree of crystallinity was determined for the PE matrix.The TG analysis was performed in ambient air and with a simultaneous TGA-DSC thermal analyser STA 449 ***F1***
*Jupiter^^®^^* manufactured by Netzsch (Selb, Germany). The TG tests were carried out at a heating rate of 10 °C/min and within a range of 30–800 °C (and 1000 °C as the maximum limit for PLA) (with an argon gas shield flow of 20 mL/min), in an open crucible made of Al_2_O_3_. The reference container was an empty Al_2_O_3_ crucible. The loss mass temperatures (*T*_1%, 5%, 10%, 50%_), peak maximum decomposition temperatures (*T*_max_), and residual mass were determined.The chemical structure analysis of the tested plastics was performed with FTIR spectroscopy. The FTIR spectra were developed by applying attenuated total (internal) reflection (ATR/FTIR) with the use of a FTIR TENSOR 27 spectrometer (Bruker, Germany), complete with a *PIKE* measuring cell which features crystalline diamond embedded in zinc selenide. The FTIR spectra were collected within a range of 4000–600 cm^−1^, with 32 scans per one test specimen, at a resolution of 4 cm^−1^. An absorption mode was used for these measurements. The ATF/FTIR test specimens were the test plastics in granulated or thin film form.MFR (melt flow rate) was determined on the granulated form of the processed test plastics. The mass flow rate was tested with a MeltFlow TQ6841 load plastometer manufactured by Ceast (Turin, Italy) and with the test method from ISO 1133 [61].Determination of the geometrical features in the sampled test strips of film, which included: layflat film width, single-ply film thickness, test strip length, and blow-up ratio (ratio of the diameter of a blown film bubble (at its largest point) to the diameter of the extrusion die it comes out of), and draw down ratio (the ratio of die opening thickness to product thickness);Determination (by measurement or calculation) of the blown film extrusion process defining parameters, including: windup roller rotational speed, plasticized material extrusion speed, film haul-off velocity, test strip mass, and mass and volume flow rates;Determination of normal density with the immersion method from ISO 1183-1A [62];

## 3. Results

### 3.1. Differential Scanning Calorimetry

The DSC analysis of the tested plastic granulates revealed that all materials had a partially crystalline structure, as evident from the endothermic peaks mapped to the crystalline phase melting (Figure 1). A study of the DSC results (Table 2) revealed that the bioplastics featured many additives.

The values of *T_g_* (below −110 °C) revealed that LDPE [36] was one of the additives, and a polymer that is not biodegradable. The values of *T_g_* were approximately at −30 °C as read from the DSC curves plotted for PLA, TPS-C, and TPS-P, and partial to presence of other processing additives in the tested bioplastics. Reference literature suggests it could include PBS (poly(butylenesuccinate)) [20,21,23,25]. Addition of PBS to a polymer (like PLA) increases the degree of crystallinity and flexibility without reducing biodegradability [20,21]. The presence of pure PLA component in the bioplastic was supported by its determined *T_g_* (at around 61–65 °C in amorphous D-lactide isomer) and *T_m_* at approximately 147 °C (for partially crystalline L-lactide isomer) [19,21,48,63,64,65]. For the first heating cycle of PLA, the DSC curve revealed glass transition of LDPE, PBS and PLA (in this order), followed by melting of LDPE and PBS (*T_m_* at about 121 °C), and finally, melting of PLA.

The DSC curves plotted for the second heating cycle, no PLA melting peak was present; among all plastic components, PLA is least prone to crystallization. For the plastics with TPS, the DSC curve in the first heating cycle was very much like the DSC curve of PLA. This suggests that the plastics could share similar additives. However, a very wide endothermic peak of TPS melting (*T_m_* at about 124 °C) was superimposed on the LDPE melting peak (*T_m_* at slightly below 120 °C). The similar values of *T_m_* for the starch-based materials were provided by other researchers [40,48,53]. The DSC curves plotted for the second heating cycle of the biodegradable plastics show much lower endothermic peaks than in the first heating cycle. This would suggest a worse capacity for crystallization in the biodegradable plastics. As mentioned in the introduction to this paper, native starch requires pre-treatment by modification to enable downstream processing. The treatment is based on the application of various plasticizers or the addition of native starch to non-biodegradable plastics, like PE, to formulate polymer blends [28,33,36,54,66,67,68,69,70,71]. The results of the DSC analysis allow a conclusion that this type of modification/application of starch as a biodegradable polymer in production of thermoplastic bioplastics was used in commercial plastics, including BIOPLAST GF 106/02 and BioComp^®^BF 01HP.

The test results, developed from the DSC curves of plastic film types, are shown in Table 3. An analysis of the test results showed that the blown film extrusion process increased the degree of crystallinity in the products. This could be attributed to an improved organisation of the film structure in comparison with the unprocessed plastic; the plasticizing with subsequent cooling of the plastic provided a better separation between the crystalline and amorphous phases. This thesis was proven by the *T_g_* values, which were lower in the extrusion-blown film than in the input granulates of the plastics. An increase in the degree of crystallinity in the film, from the input granulate level, could also be caused by a two-axial extension during the blown film extrusion process; this was noticed by Liu et al. [72] and elaborated on by Osborn [73]. For the LDPE film, the increase of the extruder screw rotational speed reduced the degree of crystallinity in the produced film; none of the tested bioplastics revealed this relationship. In all measurement series, the exit velocity had no significant effect on the physical transition temperature. In some instances, there were significant differences in Δ*H.*

### 3.2. Thermogravimetric Analysis

Table 4 lists the most important data from the TGA of the plastics tested in the granulated form, whereas Figure 2 shows the TG and DTG curves. The results show that all tested plastics remained stable in air up to 250 °C, with the TPS-C sample being least thermally stable. Unlike the TG results for the respective plastic films, the TGA of the input materials did not reveal a relatively high moisture (water) content (which are significant to processability). Whereas, the FTIR spectra for TPS-C and TPS-P revealed that water was present in the plastics. Aside from water, the plastics could release plasticizers (if any) at the onset of thermal decomposition [23,39,48]. The determined *T*_max_ values of the tested plastics were represented in reference literature. The references concerning thermal decomposition of PLA specify that the maximum decomposition rate of the material is between 266 and 376 °C [24,25,48], depending on the quantity of the isomers suitable for the process. For starch-based materials, thermal decomposition of starch begins already at 300 °C [23]. The authors of [48] find that for the PLA/TPS blends, the starch decomposed before PLA. This suggests that PLA should have a better thermal stability than starch-based plastics. The results provided here by the authors satisfied the assumption. The presence of a filler in PLA was at very similar levels in the pure plastic and the extrusion-blown film (with the post-decomposition residual mass at 14% in the crucible). The values of *T*_max_ read from the DTG curves are partial to complexity of the tested plastics (which means bonds of varying chemical stability were present in the materials), originating from the processing additives and the chemical structures of the plastics. This corroborates the conclusions made from the DSC results.

Table 5 lists the most important data outputs of the TGA of the plastic films. These results can lead to a conclusion that the processing did not affect thermal stability of the produced plastic films. The plastic films contained water (between 1.5% to 3% of the specimen mass, see the respective curves). The prior FTIR test also revealed water in the test specimens. An analysis of the exit velocity on the thermal stability of the extrusion blown film allows a conclusion that the increase of exit velocity caused a slight growth in thermal stability in the PLA series. The relationship was not this obvious in the test series for other plastics.

### 3.3. Chemical Structure

Figure 3 shows the ATR/FTIR spectra for the granulated form of the tested plastics. The granulated LDPE spectrum reveals the absorption bands characteristic of the C–H vibration in CH_2_ groups (developed at approximately 2915, 2848, 1468, 1376, and 719 cm^−1^). The granulated PLA spectra reveal the absorption bands characteristic of C–H valence vibration in CH_2_ and CH_3_ groups (between 2953 and 2849 cm^−1^) and C–H bending vibration at 1455 and 1360 cm^−1^. The absorption bands around 1713 cm^−1^ were partial to the presence of carbonyl groups of the ester group in PLA, which was additionally proven by the absorption bands within 1119–1101 cm^−1^ (C–O–C group vibration) for the PLA structure, whereas the band at approximately 870 cm^−1^ was partial to the presence of C–C bonds [20,25,48,70]. Furthermore, the low-intensity bands revealed the presence of additives in the tested plastic, as confirmed in prior by DSC and TG. The FTIR spectra of the granulated TPS-C and TPS-P did not vary greatly between the two. The only noticeable difference in those FTIR spectra is the band at 1646 cm^−1^ and a band at about 3200 cm^−1^ and a narrow band at 3395 cm^−1^ superimposing the broad peak at 3500 cm^−1^, which was more prominent for TPS-C. Bands at about 1646 and 3200 cm^−1^ could indicate a presence of an amides, which can be often used as antistatic or antiblocking agents in plastics. In addition, there are differences in amylose and amylopectin content for the starch based plastics tested but this is not visible on these FTIR spectra. Aside from the evidence of CH_2_, CH_3_, C=O, C–O, and C–O–C group bands, there was a distinct band around 3395 cm^−1^ and characteristic of hydroxyl groups (C–OH), and it was more intensive than in the PLA film. This could suggest, among other things, a small amount of water that were present in the starch-based test specimens, or a plasticizer, like glycerol [38,39,48,54].

When studying the effect of varying screw rotational speed on the chemical structure of the films produced from the tested plastics, an explicit conclusion can be drawn that within each of the film measurement series (with each produced at a different exit velocity) no distinct or significant changes occurred in the FTIR spectral images.

### 3.4. Melt Flow Rate

The results determined for the melt flow rate at 190 °C, at the blown film extrusion temperatures of the tested plastics, and under a test load of 5 kg, are shown in the Figure 4.

A study of the results reveals that the highest MFR was found in PLA (BioComp^®^BF 7210) processed at 190 °C. The lowest MFR was found in TPS-P (BIOPLAST GF 106/02). The difference between the two values was unusually high, TPS-P would flow 4.5 times worse than PLA. A lower processing temperature distinctively reduced MFR. When the temperature was increased from 145 °C to 190 °C, MFR in the BioComp^®^BF 7210 PLA increased 6-fold. MFR in the BIOPLAST GF 106/02 TPS-P increased more than two times when the temperature changed from 160 °C to 190 °C. A processing temperature change in the BioComp^®^BF 01HP TPS-C from 155 °C to 190 °C provided a three-fold increase in MFR. The FABS 23-D022 LDPE behaved somewhat similar to TPS-P, with MFR which grew over two times from 160 °C to 190 °C.

There was a noticeable and very large difference between TPS-P and TPS-C at 190 °C and 160 °C. This was undoubtedly caused by the presence of amylose macromolecules (15–30%) which have a linear structure and amylopectin (70–85%), the macromolecules of which are branched. The level of amylopectin depended on the origin of native starch and had a significant impact on the degree of crystallinity and the supermolecular structure of starch [41,42]. In potato starch, the level of amylopectin was much higher; hence the lower MFR. However, in our case, a lower MFR may be associated more with a higher molecular weight than with a higher content of amylopectin [44].

### 3.5. Geometric Features

Figure 5 and Figure 6 reveal that the geometric features of the produced film (Figure 7), i.e., the layflat film width and the single-ply film thickness were within the limits predefined in the test plan. The differences discovered are a result of various disturbance factors present in the blown film extrusion process. The test strip length (Figure 8) sampled in 15 s intervals increased with the screw rpms of the extruder in each of the extruded plastic material. It was found, however, that for the present layflat film width and single-ply film thickness, the test strips sampled from the biodegradable plastics were definitely longer than those from LDPE. This would suggest that potential differences existed in the flowing mechanism between the specific plastic types and in the film sleeve blow-forming, and that it would be necessary to individually adjust the remaining blown film extrusion parameters to produce film with the required layflat film width and single-ply film thickness.

### 3.6. Blown Film Extrusion Characteristics

One of the processing parameters of blown film extrusion applied in this research was the rotational speed of film windup rollers. The trends of change in the windup roller rpms for each of the test-processed plastic were closely correlated to the produced test strip length. Figure 9 shows a markedly reduced windup roller rpms during LDPE blown film extrusion in comparison to the three biodegradable plastics. When extruding LDPE blown film and increasing the screw speed by 100 rpms, it would be necessary to increase the windup roller speed by 5 rpms on the average, to maintain a constant film thickness. For the film extrusion blown from TPS-P, TPS-C and PLA, it would be necessary to increase the windup roller speed by 10 rpms on the average to maintain a constant film thickness.

Despite the necessity to apply different windup roller speeds, the determined blow-up ratio and draw down ratio (Figure 10, and Figure 11, respectively) were very similar in all four tested plastics. The lowest blow-up ratio was equal to 3.74 in TPS-P at the screw rotational speed of 300 rpms; the highest blow-up ratio was 3.87 in PLA at the screw rotational speed of 500 rpms. The difference between these two limit values was 3.36%. The blow-up ratio depended on the blow-up ratio of the film sleeve, which grew until the required layflat film width was achieved. Hence, the trends of blow-up ratio values were consistent with the changes in layflat film width. The draw down ratio values for all tested plastics were between 10.0 and 11.5 except for PLA extrusion blown at 400 rpms of the screw, where the determined draw down ratio was 12.86. This was related to a lower film thickness in the same measurement series. The differences in the draw down ratio between the plastics could be a result from the change in the windup roller diameter. The windup roller diameter grew at a steady rotational speed as the winding up time passed and the length of wound up film was longer; this resulted in a slow increase of the film haul-off velocity. The phenomenon possibly contributed to the variations in film thickness shown in Figure 6.

The mass flow rate analysis could be performed with a quantitative approach, whereby measurement of the extent that a finished product can be produced per a specific unit of time, is carried out. A mass approach can also be applied, by which it is measured how much input material can be processed into a finished product per a specific unit of time. The quantitative parameters which defined the length of extrusion blown film were the exit velocity (the extrusion rate) (Figure 12) and the film haul-off velocity (Figure 13). The exit velocity specifies how fast the film the width of which equals that of the extrusion nozzle slot (1 mm) leaves the extrusion head. The exit velocity is directly related to the capacity of the processed plastic to flow at the specified processing conditions and the gravity feeding of granulated input material to the extruder hopper. Figure 12 shows that the best flowing plastic was TPS-P, followed by PLA, and LDPE with the poorest flowability: At 500 rpms of screw speed, the LDPE flow was approximately 1/3 worse than in TPS-P. However, these findings should be indicative only. In order to precisely understand the flowability of each of the tested plastics, MFR (melt flow rate) would have to be determined.

A potential reason for discrepancies in flow velocity and melt flow rates at processing temperatures may be the effect of shear resulting from the rotational motion of the screw, especially since the rotational speeds used were quite high. The MFR charts show a significant increase in the melt flow rate of biodegradable materials with increasing temperature, in particular in the case of PLA, which at 190 °C had a very good MFR. The shear effect can cause an autothermal effect, whereby the local material mass temperature can be much higher than the current barrel wall temperature, which can potentially result in even a drastic increase in melt flow rate. Measurement of the rate allows you to get an idea of how the processability of materials is shaped. However, due to the static nature of the measurement, it does not fully reflect what is happening in the plasticizing system. The effect of processing on the physical properties of materials is confirmed by the analysis of DSC results of the first heating cycle for granules and films. The extrusion process did not affect the thermal properties of PE at all, the values of *T*_g_, *T*_m_, Δ*H* and *X* for the granulate and the obtained films are comparable. Clear differences can be seen for materials based on potato and corn starch. The TPS-C film has twice higher ΔH and X in relation to its granulate, and the TPS-P film three times. This proves the significant impact of processing on the properties of these materials.

The film exit velocity affected directly the mass flow rate of the blow film extrusion process. If a plastic material exited the extrusion head faster, more film could be produced in a unit of time. Figure 12 and Figure 13 reveal that the exit velocity of TPS-P, TPS-C, and PLA provided more completed film extrusion-blown, per hour, than LDPE. For all tested materials and screw rpms, approximately 10 m of film could be produced from one metre of blown-up sleeve, which was consistent with the draw down ratio values shown in Figure 10. Note that a screw rotational speed of 500 rpm could provide almost twice as much biodegradable plastic film than the traditional LDPE packaging film.

The blown film extrusion mass flow rate analysis with the processed input material mass required measurement of the mass of the test strips (Figure 14) sampled every 15 s. The lowest average test strip mass was found in LDPE, which was consistent with the previous findings. However, the test strip mass would not only depend on processability of plastic, it also depended on density of the material. Note that all test strip mass measurements had much lower error margin than the test strip length measurement, which resulted in lower standard deviation values.

Figure 15 reveals that the mass flow rate vs. screw rpms had a linear increase of approximately 6 kg/h in the biodegradable plastics for each 100 rpm increment of screw rotational speed. In LDPE, the mass flow rate increment was also directly proportional to the screw rotational speed, but it was only approximately 1.5 kg/h per every 100 rpm of screw rotational speed increment. As a result, for the highest screw rpms tested, the LDPE processing mass flow rate was 16 kg/h (57%) lower than in the tested biodegradable plastics. This obviously affected the energy efficiency of blown film extrusion [37,74].

The calculated volume flow rate vs. screw rotational speed (Figure 16) followed a trend similar to that in the mass flow rate tests. Note, however, that the difference in volume flow rate between LDPE and the biodegradable plastics was much lower than in mass flow rate. For the highest screw rotational speed tested, the volume mass flow rate of LDPE was lower by 34% from that of the biodegradable plastics. Based on this, LDPE had a lower density than TPS-P, TPS-C or PLA. By analogy, a conclusion can be made that TPS-P was less dense than TPS-C, since TPS-P had lower mass flow rate and a higher volume flow rate than TPS-C. The normal density values determined in the immersion test confirmed this assumption. The average normal density values were as follows: PE at 925.3 kg/m^3^, TPS-P at 1214.4 kg/m^3^, TPS-C at 1221.2 kg/m^3^, and PLA at 1290.9 kg/m^3^.

## 4. Conclusions

The tests in blown film extrusion of the four different polymer materials, LDPE, commercial name Malen E FABS 23-D022; potato starch based plastic (TPS-P), BIOPLAST GF 106/02; corn starch based plastic (TPS-C), BioComp^®^BF 01HP; and a polylactic acid (polylactide) plastic (PLA), BioComp^®^BF 7210, revealed that the biodegradable plastics sources, from renewable material sources, could be very efficiently processed with traditional extrusion machines, and specifically:The error bars, obtained for the geometrical features of the extruded film, were clearly higher for biodegradable materials than for LDPE. This can be interpreted as a lower stability of the extrusion process and a greater susceptibility to interference from PLA-based materials and starch. DSC tests showed significantly lower values of melting heat for biodegradable materials, which may be the basis for finding greater susceptibility to temporary changes in physical properties, caused by a slight decrease in the temperature of the bubble. Such temperature fluctuations can be caused, for example, by air blasts, especially since the film extrusion process was carried out in an open production hall of considerable size. Obtaining high dimensional repeatability for biodegradable materials tested may therefore require ensuring more controlled conditions.At a screw rotational speed of 500 rpm could provide almost twice as much biodegradable plastic film than the traditional LDPE packaging film.With the screw rpms increasing, the exit velocity of blown film extrusion from the biodegradable plastics would grow more intensely than in the LPDE film. The retention of comparable film thickness values, draw down ratios and blow-up ratios with the increasing speed rotational speed increased the exit velocity twice in the biodegradable plastics when compared to LDPE.The DSC analysis revealed that the processing by blown film extrusion changed the crystalline structure between the input granulate and the produced film. However, the chemical structure characterized by FTIR spectroscopy did not change. The variation of screw rotational speed did not significantly affect the thermal properties of any of the produced films, which is an important insight for the determination of the blown film extrusion process parameters. The TG analysis (corroborated by the FTIR spectra) revealed that the granulated PLA, TPS-C and TPS-P and the films produced from these input materials, included some amounts of water, which could cause difficulties in reprocessing.With respect to processing, the plastic most resembling LDPE was TPS-C, and PLA had a better flowability, followed by TPS-P, which was worse in this regard. It was also noted that the processability of the tested biodegradable plastics was more affected by temperature variations than LDPE. Temperature can significantly control viscosity and other rheological properties.

## Figures and Tables

**Figure 1 materials-13-01986-f001:**
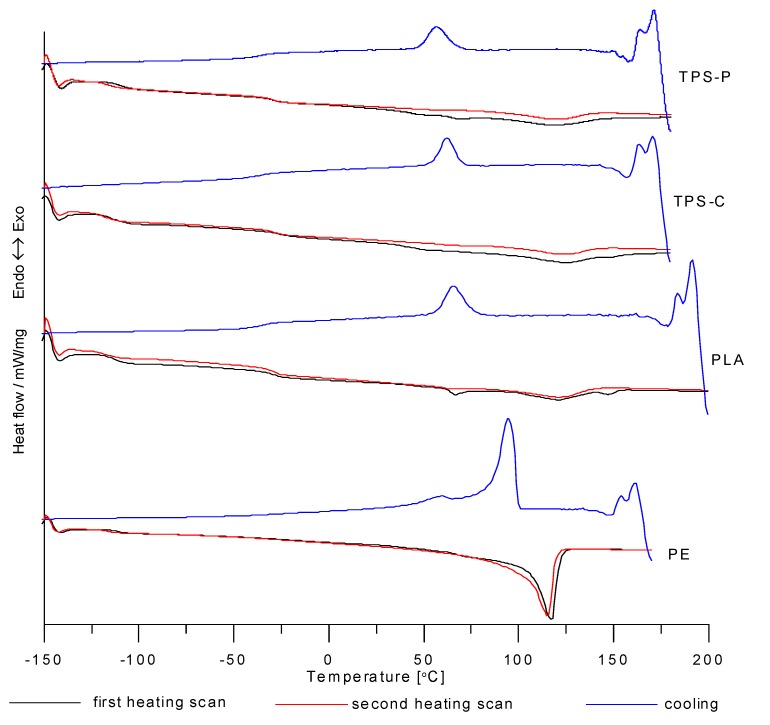
DSC curves of tested granulates.

**Figure 2 materials-13-01986-f002:**
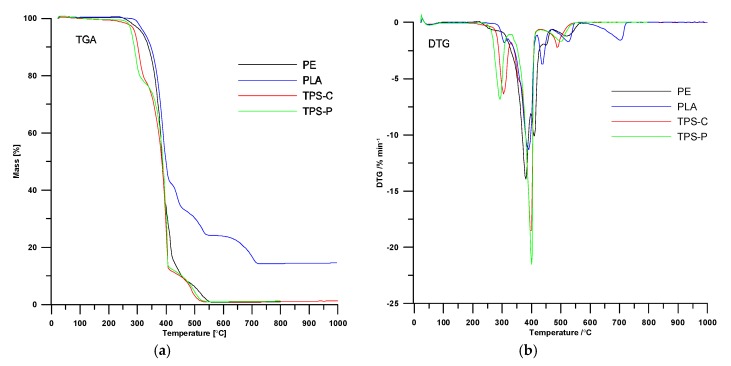
TGA (**a**) and DTG (**b**)curves of tested granulates.

**Figure 3 materials-13-01986-f003:**
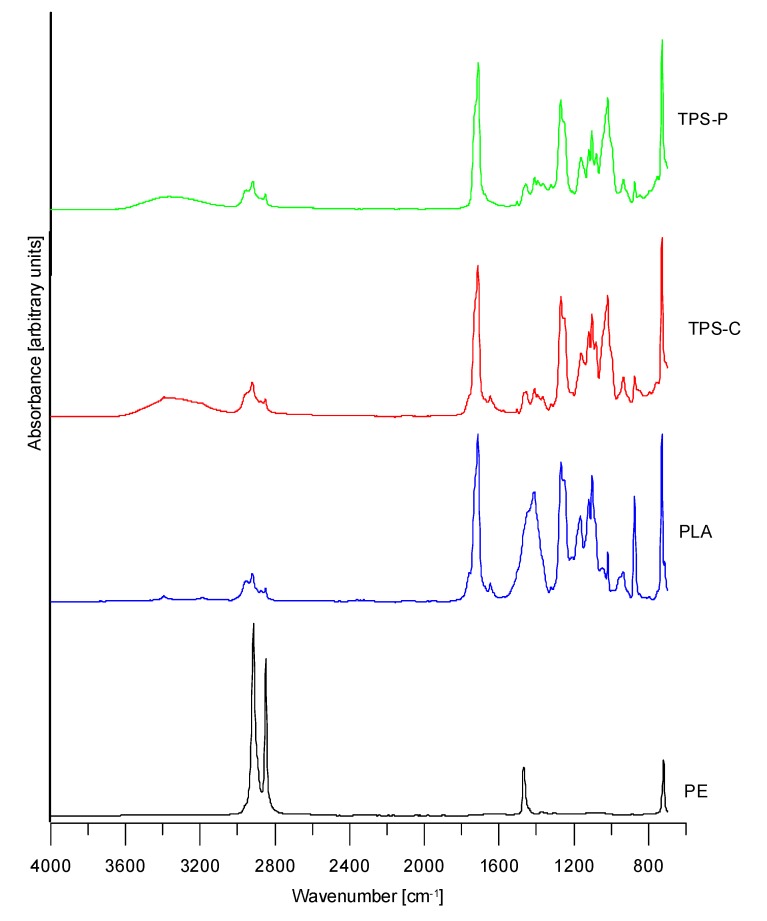
FTIR/ATR spectra of tested granulates.

**Figure 4 materials-13-01986-f004:**
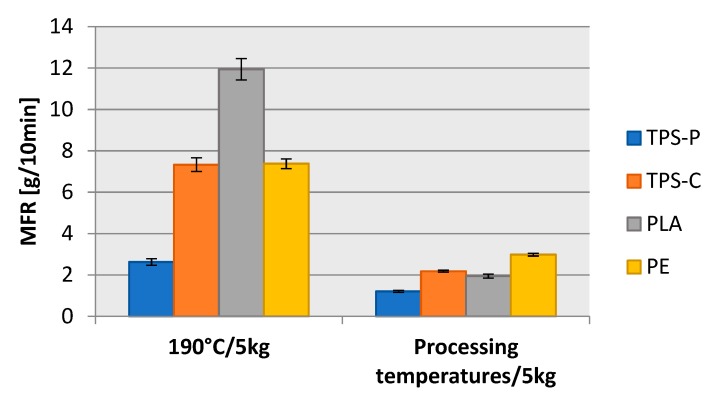
Measurements of melt flow rate of tested materials at 190 °C (cut off times: PLA and TPS-C—5 s, LDPE—10 s, TPS-P—15 s) and processing temperatures (cut off times: LDPE—15 s, PLA—30 s, TPS-C—15 s, TPS-P—30 s).

**Figure 5 materials-13-01986-f005:**
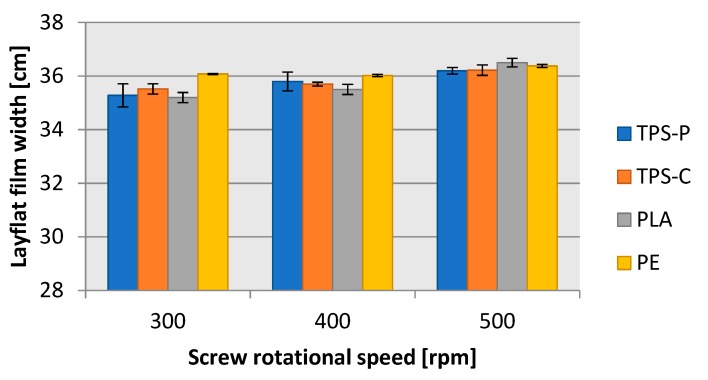
Dependence of the width of the obtained films on the rotational speed of the extruder screw.

**Figure 6 materials-13-01986-f006:**
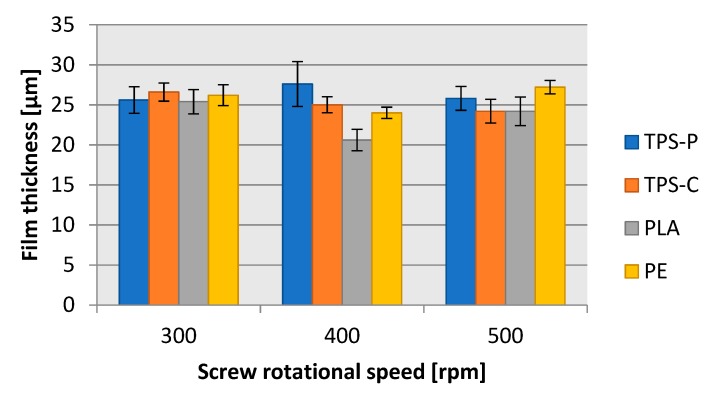
Dependence of the thickness of the obtained films on the rotational speed of the extruder screw.

**Figure 7 materials-13-01986-f007:**
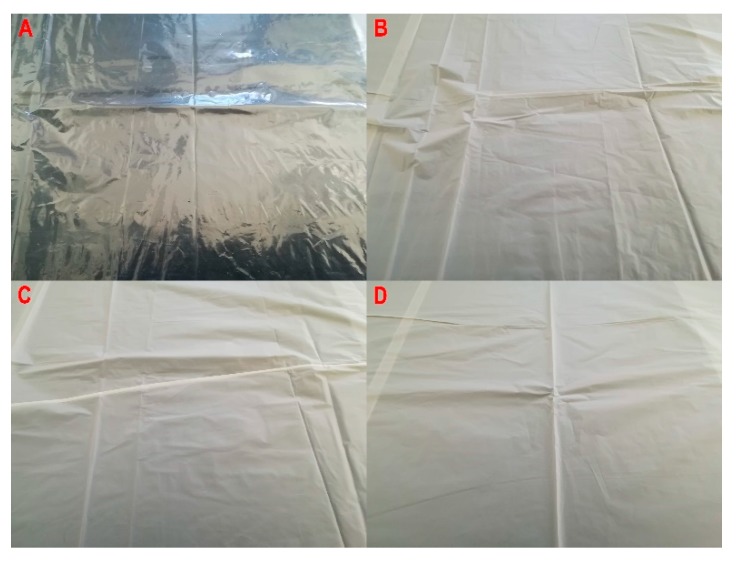
The appearance of the film made at a screw speed of 400 rpm: (**A**) LDPE, (**B**) PLA, (**C**) TPS-C, (**D**) TPS-P.

**Figure 8 materials-13-01986-f008:**
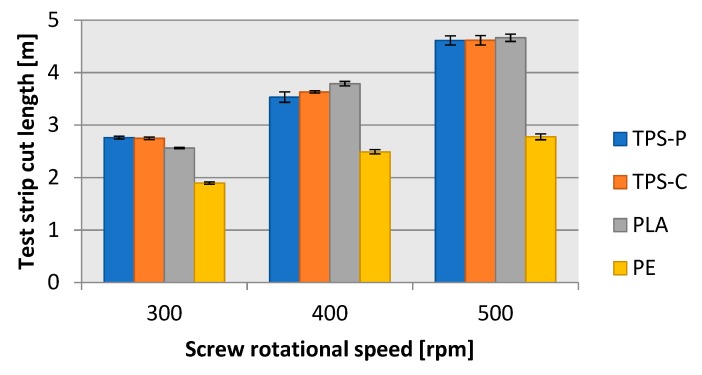
Dependence of the length of measuring sections of the obtained films on the rotational speed of the extruder screw.

**Figure 9 materials-13-01986-f009:**
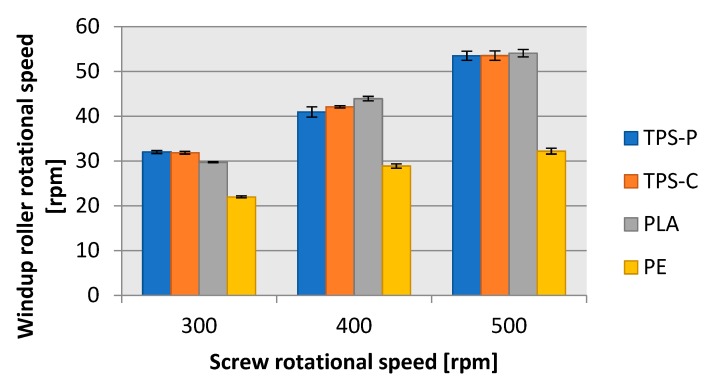
Relationship between the rotational speed of the receiving rollers and the rotational speed of the extruder screw.

**Figure 10 materials-13-01986-f010:**
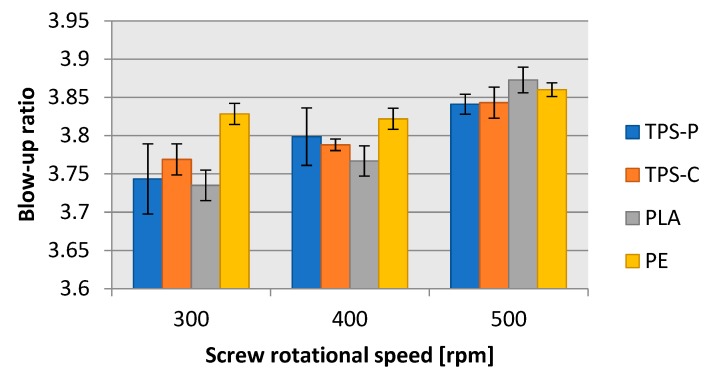
Relationship between the blow-up ratio (degree of transverse stretching) of the obtained film and the rotational speed of the extruder screw.

**Figure 11 materials-13-01986-f011:**
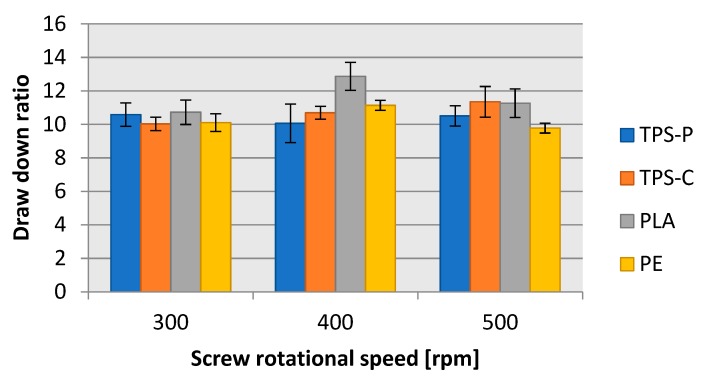
Relationship between the draw down ratio of the obtained film and the rotational speed of the extruder screw.

**Figure 12 materials-13-01986-f012:**
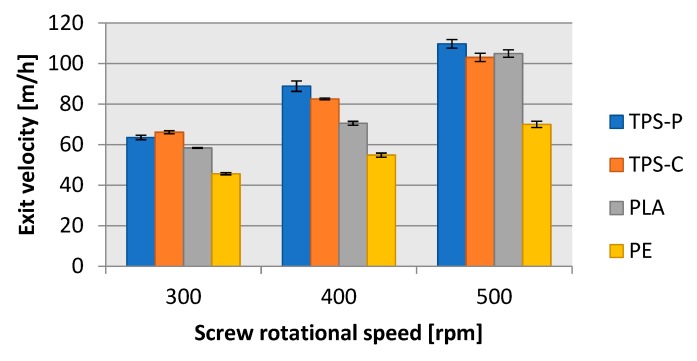
Relationship between film exit velocity on the rotational speed of the extruder screw.

**Figure 13 materials-13-01986-f013:**
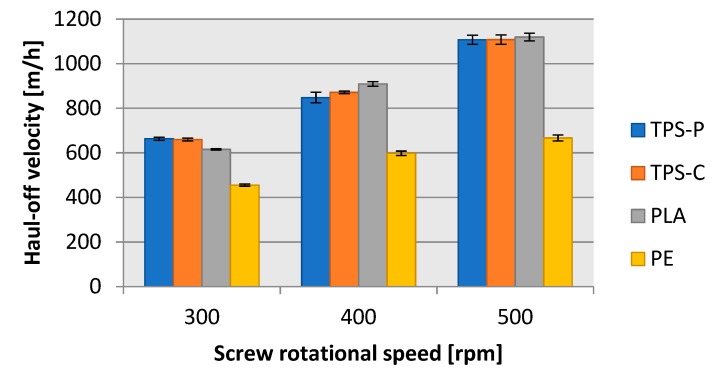
Dependence of the haul-off velocity of film collection on the rotational speed of the extruder screw.

**Figure 14 materials-13-01986-f014:**
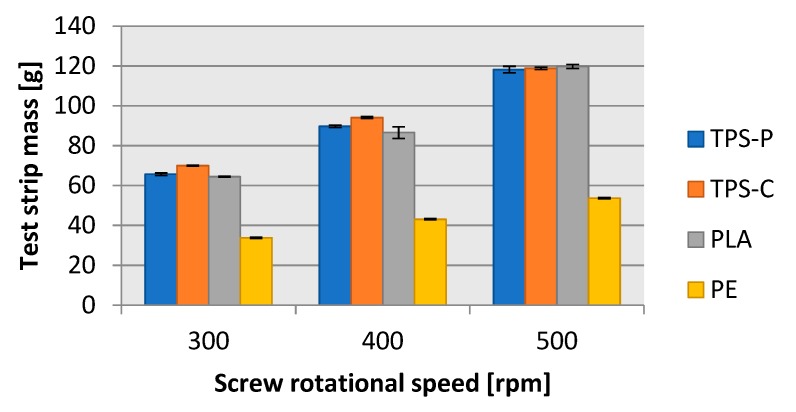
Dependence of the test strip mass of film collection on the rotational speed of the extruder screw.

**Figure 15 materials-13-01986-f015:**
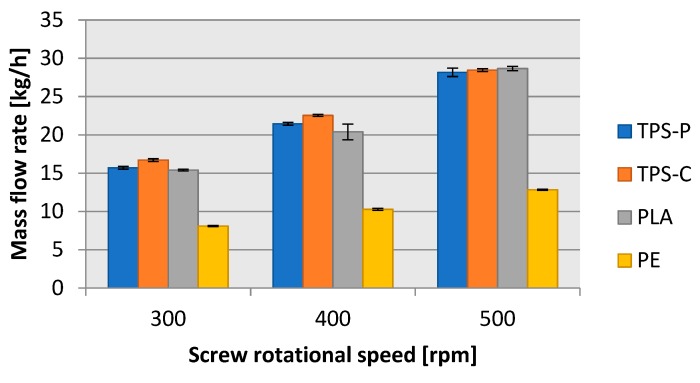
Dependence of the mass flow rate of the film measuring section on the rotational speed of the extruder screw.

**Figure 16 materials-13-01986-f016:**
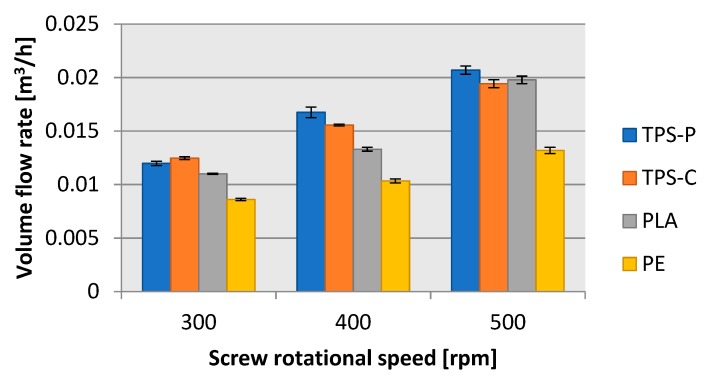
Relationship between the volume flow rate of the material and the rotational speed of the extruder screw.

**Table 1 materials-13-01986-t001:** Comparison of selected properties of tested materials and foils made of them according to manufacturers’ data [56,57,58,59].

Parameter	PE	TPS-P	TPS-C	PLA
Density, kg/m^3^	923	1200–1300	1270–1300	1380
MFR, g/10 min (190 °C, 2.16 kg)	1.95	2.5–5.5 *	2–6	10.76 *
Melting point, °C	112	120–130	110–130	140–150
Tensile strenght MD, MPa	18	20–35	18	35.7
Tensile strenght TD, MPa	17	20–35	10	25.7
Tensile elongation MD, %	450	600–900	200	250
Tensile elongation TD, %	540	600–900	250	610

* value given for a load of 5 kg, TD—transverse direction, MD—longitudinal machine direction.

**Table 2 materials-13-01986-t002:** DSC measurement results for granulates.

Specimen	*T*_g_ (°C)		*T*_m_ (°C)		Δ*H* (J/g)		X (%)	
	I	II	I	II	I	II	I	II
PE	−111	−116	117	115	156	163	53.24	55.56
PLA	−113 −31 65	−117 −29 -	121 147	121	7.4 1.12	9.38	1.08	-
TPS-C	−115 −26	−117 −29	124	124	45.7	11.7	15.63	4.03
TPS-P	−109 −31	−119 −31	120	120	36.4	12.5	12.4	4.27

where: I, II—first, and second heating cycle, respectively.

**Table 3 materials-13-01986-t003:** DSC measurement results for films.

Specimen	*T*_g_ (°C)		*T*_m_ (°C)		Δ*H* (J/g)		X (%)	
	I	II	I	II	I	II	I	II
PE 300	−116	−113	112	112	162	161	55.35	55.08
PE 400	−120	−123	112	112	157	162	53.72	55.42
PE 500	−120	−116	112	112	153	156	52.14	53.33
PLA 300	−116 −31 48	−120 −31 59	117 148	120	7.6 1.23	9.84	1.38	-
PLA 400	−120 −26 48	−118 −30 59	118 148	120	9.28 1.04	9.28	1.17	-
PLA 500	−122 −31 49	−118 −29 59	117 149	120	7.52 1.38	9.81	1.49	-
TPS-C 300	−117 −28	−120 −30	96	122	91	13.4	31.06	4.59
TPS-C 400	−122 −29	−121 −32	92	122	96.5	17.13	32.94	5.85
TPS-C 500	−113 −28	−120 −30	104	122	89.94	17.17	30.70	5.86
TPS-P 300	−114 −30	−112 −28	102	122	114.5	9.99	39.08	3.41
TPS-P 400	−115 −29	−117 −29	104	123	125.8	10.18	42.92	3.47
TPS-P 500	−114 −30	−117 −30	104	121	116.1	10.43	39.63	3.56

where: I, II—first, and second heating cycle, respectively.

**Table 4 materials-13-01986-t004:** Results obtained based on TG curves for granulates.

Specimen	*T*_1%_ (°C)	*T*_5%_ (°C)	*T*_10%_ (°C)	*T*_50%_ (°C)	*T*_max_ (°C)	Residual Mass (%)
PE	270	317	339	386	381; 410; 451; 521	0.92
PLA	299	324	348	399	307; 389; 437; 525; 703	14.54
TPS-C	232	292	302	386	305; 359; 400; 489	1.20
TPS-P	259	280	289	389	292; 401; 501	1.20

**Table 5 materials-13-01986-t005:** Results obtained based on TGA curves for films.

Specimen	*T*_1%_ (°C)	*T*_5%_ (°C)	*T*_10%_ (°C)	*T*_50%_ (°C)	*T*_max_ (°C)	Residual Mass (%)
PE-300	282	330	349	396	355; 387; 413; 426; 451; 505	1.16
PE-400	302	330	347	401	342; 360; 375; 426; 451; 463; 502; 650	0.70
PE-500	277	325	345	384	334; 373; 401; 418; 446; 511	1.36
PLA-300	298	324	348	401	309; 391; 441; 529; 702	14.79
PLA-400	302	329	349	402	312; 390; 440; 528; 703	14.82
PLA-500	307	331	349	401	390; 438; 529; 704	11.97
TPS-C 300	258	297	306	387	309; 397; 481; 640	1.33
TPS-C 400	254	295	304	386	306; 397; 481; 640	0.78
TPS-C 500	256	297	306	389	308; 399; 485; 640	1.15
TPS-P 300	269	286	295	392	294; 398; 507	0.98
TPS-P 400	273	287	296	390	296; 397; 440; 480; 530; 665	0.96
TPS-P 500	271	289	298	392	299;398; 443; 487; 532; 665	1.19

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
