# Peer review of "Modern Biodegradable Plastics—Processing and Properties: Part I"

_materials, 2020, doi:10.3390/ma13081986_

Round 1

Reviewer 1 Report

The authors provide significant processing and thermal stability of a series of biopolymers compared to LDPE for film applications.  The work shows that all of the polymers are processable and reasonably stable. The authors also conclude that each compositions contains polyethylene. The premise of the work is that the biopolymers are ecologically good - yet, history has taught us (recall the Glad(R) trash bags made of PE and starch) that combining PE with a biodegradable component simply makes the PE more bioavailable - i.e. more hazardous than unmodified PE. The authors ignore this critical point.  The authors provide FTIR data for compositions, but the results are not compelling.  Something more substantial such as NMR probably is needed here.  Page 4 line 159 should read talc not talk. Table 1 is in Polish.  Page 10/384 should say "supported by" not "proven" as this is clearly short of proof. There are no mechanical properties reported, so what good is this study to someone interested in these materials. Yes, they are fairly stable to heat, but are they strong? Ductile?  Have moisture and/or oxygen barrier?  The results are at best a processing comparison and as such should be re-written for a processing journal.

Author Response

Dear Reviewer

Thank you for commenting on the article.

I think that the implemented changes will make the manuscript more clear and scientifically transparent. Most of the changes in the text are presented in file "author-coverletter-6869441.v1.docx" in color. Referring to the reviewer's comments, we would like to note that toxicity tests were not the subject of the article, but the materials selected for testing have compostability certificates. According to the description from EN 13432 and EN 14995, compostable material is one that is biodegradable under aerobic conditions in an amount of 90%, over a period of 6 months, and the products resulting from this decomposition should not adversely affect the quality of the compost, and consequently also the environment. NMR analysis would probably be useful, but many examples can be given, even from the magazine "Materials", that FTIR research is sufficient, however, to take into account the reviewer's suggestion, FTIR analysis has been completed and extended.

Talk in line 159 was corrected to talc.

Table 1 was translated to English.

On page 10/384 proven was replaced by supported.

According to the manuscript title, the content of the work corresponds to it. The authors decided to split the research performed into two articles due to the large amount of content. The first part contains the processing characteristics and the impact of the processing process on the thermal properties and chemical structure of the materials tested. The second part will contain the results of tests of mechanical and tribological properties, microscopic structure analysis, surface roughness analysis, color and gloss tests as well as results of barrier properties tests. The authors considered the division of results into the characteristics of the production process (part I) and the characteristics of the properties of the obtained films (part II) as logical. We would like to say that the second part is very advanced and will be ready to be sent to the editor by the end of May 2020 at the latest.

We hope that our answers will remove any doubts as to the publication of this manuscript.

Best regards,

Janusz Sikora

Reviewer 2 Report

The paper is about the film blowing of selected biodegradable plastics and a polyethylene grade. The objective is to provide a comparative study of plastics processability and of film properties, targeting the packaging industry as an application. In spite of the very bright interest in polymer science and engineering for the topic of the paper, its publication would be premature at this stage.

Results about the film blowing process are presented before the characterization of materials, which is at odd with a logically grounded study. Namely, materials should be first characterized in order to better set the processing parameters which will permit a satisfactory film blowing.

Then, the idea of this study is quite strange. What is the reason for comparing the rate of production of films? Why is the study not focusing on the processability of each material, by characterizing the operability window of each material? One thing is to blow films at a certain production rate, the other is to produce films with a stable process which can run hours to guarantee satisfactory film properties (homogenous width and thickness, no wrinkles, etc.). From the set of data on films “geometrical features”, it seems clear that error bars are much smaller for LDPE films, which suggest that the process is stable for this material. How is bubble stability for the others? I thus invite the authors to carry a processability study as for instance in Mallet, Lamnawar and Maazouz (2014) focusing on the range of blow up ratio and take up ratio that give stable bubbles.

Coming back to the production rate, why choosing a specific set of processing parameters which definitely do not favor the results obtained with LDPE? In other words, targeting the production of films with defined lay flat width and thickness is sounded. But establishing a single processing temperature for each material with no further explanation than “manufacturer’s data and proprietary experience of the authors” is not scientifically acceptable. More important, authors claim that LDPE poor flowability is responsible for the smaller production rate. But increasing the processing temperature would have solved the problem. LDPE is known to be a winner for film blowability, as the window of processing temperatures (not mentioning other processing parameters) giving satisfactory films is huge in comparison with the selected biodegradable plastics. Moreover, for this specific process, flowability is a much less important rheological attribute than melt strength. So, as far as materials characterization is concerned, a rheological study incorporating extensional viscosity or at least elasticity will be required.

Finally, since packaging application is targeted, films characteristics which are of peculiar interest for such application should be reported. These are for instance optical properties, tear resistance, barrier and welding properties. Such characteristics could be reported in this paper at no cost for its length as many figures and data are redundant (for instance there is no need to report windup roller speed and haul-off velocity as the draw down ratio incorporates such data; volume flow rate, exit velocity, mass flow rate, test strip mass and length, since all are related, etc.), or can be collapsed in a single table (for instance DSC data for granules and films)., or simply do not need to be reported (e.g. experimental data from computing MFR values, which on the contrary do require the computation and report of error bars).

Author Response

Dear Reviewer

Thank you for commenting on the article.

I think that the changes made will make the manuscript more clear and scientifically transparent. Most of the changes in the text are presented in file "author-coverletter-6874850.v1.docx" in color.

As suggested by the reviewer, the order of the results was changed so that the results characterizing the materials used for the research go first, and they are followed by the results about the film blowing process.

The reviewer is generally right in their comments, but our goal was not to carry a processability study focusing on determining the range of blow up ratio and take up ratio that give stable bubbles nor on focusing on the processability of each material, by characterizing the operability window of each material. We wanted to present primarily the properties of films obtained from various materials and determine their suitability for applications in the packaging industry. The second part of the manuscript will contain the results of tests listed by the reviewer on the properties of the obtained films. In this manuscript, we have shown that using standard machines at film manufacturers, plastics from renewable sources can be processed, and the presented processing parameter sets are exemplary, not yet perfect. Our goal was to obtain a film with a flattened sleeve width of 35-36cm and a single layer thickness of 20-25µm, and the rotational speed of the extruder screw was a variable factor (this is described in the manuscript under "Research program and methodology").

We are aware that LDPE will be the winner for film blowability, as the window of processing temperatures giving satisfactory films is huge in comparison with the selected biodegradable plastics but we wanted to compare and see if it is possible at all to obtain films with comparable geometric characteristics (thickness and width) ) from LDPE and biodegradable plastics and whether the films obtained have acceptable properties for packaging applications (II part). This approach to the problem is useful and interesting primarily for entrepreneurs, although we are aware that there may be different points of view and approaches to the problem. 
If the reviewer's opinion is that the phrase "proprietary experience of the authors" is not appropriate, it has been removed from the manuscript.

According to our manuscript title, the content of the work corresponds to it. The authors decided to split the research into two articles due to the large amount of content. The first part contains the processing characteristics and the impact of the processing process on the thermal properties and chemical structure of the materials tested. The second part will contain the results of tests of mechanical and tribological properties, microscopic structure analysis, surface roughness analysis, color and gloss tests as well as results of barrier properties tests. The authors considered the division of results into the characteristics of the production process (part I) and the characteristics of the properties of the obtained films (part II) as logical. We would like to say that the second part is very advanced and will be ready to be sent to the editor at the end of May 2020 at the latest. The reviewer's note that some of the data given is unnecessary is not entirely correct in this case, because we refer to the technological conditions of the film blow process. The parameters listed in the article mentioned by the reviewer are indeed interrelated, but their presentation is often desired or even expected by readers, especially those dealing with the field of plastics processing, which facilitates the analysis of the issue and gives a direct view of the situation.

We hope that our answers will remove any doubts as to the publication of this article.

Best regards,

Janusz Sikora

Reviewer 3 Report

This paper can be of high interest for the readers. It is reasonably well written and clearly presented. However, iI think that some improvements are needed.

For example a figure representing the blow  extrusion equipment and explaing the geometrical features of the extruded film and the processing parameters would be of help to better understanding the results.

I have some doubts concerning the sample characterization (FTIR, TGA and MFR) that are reported as notes in the attached paper

Table 1 is still in polish, sample must be separated  and realigned in table 2, table 6 and 7 can be shortened and jointed.

There are also some typing mistakes that are also highlighted in the attache file

Author Response

Dear Reviewer,

Thank you for commenting on the article.

Most of the changes in the text are presented in file "author-coverletter-6907686.v1.docx" in color.

The reviewer suggests placing figures representing the blow extrusion equipment and explaining the geometrical features of the extruded film and the processing parameters in the article. However, we believe that this is not necessary in this case. The purpose of our work is not to study the impact of changing the geometrical parameters of the screw or processing parameters on the properties of the obtained films, but testing with the specified parameters mentioned. We also think that placing the drawing representing the blow extrusion equipment and explaining the geometrical features of the extruded film is too academic and widely known among specialists. We hope that the doubts regarding the sample characterization (FTIR, TGA and MFR) expressed by the reviewer have been clarified by us through appropriate changes and extensions of the entries in the manuscript text in response to the reviewer's notes in the text. We have completed and extended the FTIR and TGA analysis. We have improved the MFR description as suggested by the reviewer.

Table 1 has been translated into English.

Samples have been separated and realigned in table 2.

Tables 6 and 7 have been shortened and combined as suggested by the reviewer.

The typing mistakes highlighted in the attachment file have been corrected.

We hope that our answers will remove any doubts as to the publication of this article.

Best regards,

Janusz Sikora

Round 2

Reviewer 1 Report

The revisions and explanations are acceptable to me and I can support the publication of part 1.  part 2 with the mechanical data will be interesting.

Author Response

Dear Reviewer,

We would like to thank you for your effort in performing a very thorough analysis of the article in terms of its scientific aspects and style. We would also like to thank for all constructive comments and suggestions as well as for seeing potential in our work. We have revised the whole manuscript according to your constructive suggestions, which has greatly improved the quality and the presentation of the paper. Below, please find our responses to all your remarks/doubts.

Best regards,

Authors

Reviewer 2 Report

I invite the authors to include parts of the response to my comments in the last section of the introduction. This will highly clarify the purpose of the study, as in my opinion the title is too general. It does not mention PLA and TPS, whereas many other biodegradable plastics are in the market. It does not mention film blowing, and the word "processing" is misleading. Indeed, the effect of screw speed has been investigated, which is one in many other parameters.

The comment on films homogeneity (error bars), which relates to the stability of the process and impacts on e.g. film mechanical/optical/barrier properties, still needs to be addressed (see also below). This will highlight some shortcomings of PLA and TPS. Also pictures of produced films should be provided, as this is a clear indicator of “their suitability for applications in the packaging industry”.

Though the new presentation of MFR results is far more instructive, the caption to Figure 4 is wrong, as there is no screw speed effect. The discussion of MFR results does not connect to the discussion of extrusion rate and flowability. This is critically missing, as my previous comment on LDPE flowability is not addressed in the revised paper. In the abstract, it should be made clear that results for films were obtained under specific screw speed and temperatures conditions. This is because conclusions could be drastically different for other processing conditions.

Other comments:

-line 71 “extremely good processability”: this statement for PLA is at odd with the numerous literature on PLA modification for improving its film blowing ability, see for instance Kharrat, Chaari et al, J. Renewable Materials 2019. So these lines should be revised, separating PLA thermal and mechanical properties, from its processability in film blowing. In other words, it should be made clear that blowing PLA into a film is impossible, unless properly modified using different strategies.

-Figures 5 and 6: how are error bars computed? Are they computed from spatial variations within a single test strip? This also connect to a previous comment on bubble stability: since error bars are significantly smaller for LDPE, this means that the corresponding bubble and thus process is much more stable. This needs to be stated.

-Figure 8, vertical axis: units are wrong for a speed.

-line 688, “flowability”: how does flowability relate with MFR of LDPE at processing temperatures? The comment on LDPE flowability seems at odd with MFR values which are the largest.

Author Response

Dear Reviewer,

We would like to thank you for your effort in performing a very thorough analysis of the article in terms of its scientific aspects and style. We would also like to thank for all constructive comments and suggestions as well as for seeing potential in our work. We have revised the whole manuscript according to your constructive suggestions, which has greatly improved the quality and the presentation of the paper. Below, please find our responses to all your remarks/doubts.

Remark # 1: “I invite the authors to include parts of the response to my comments in the last section of the introduction. This will highly clarify the purpose of the study, as in my opinion the title is too general. It does not mention PLA and TPS, whereas many other biodegradable plastics are in the market. It does not mention film blowing, and the word "processing" is misleading. Indeed, the effect of screw speed has been investigated, which is one in many other parameters.”

Authors: The last paragraph of the introduction has been rewritten to include all the information indicated by the reviewer - the indication of the materials used, the processing technology, the processing characteristics as a test target and the screw speed as a variable factor.

Remark # 2: “The comment on films homogeneity (error bars), which relates to the stability of the process and impacts on e.g. film mechanical/optical/barrier properties, still needs to be addressed (see also below). This will highlight some shortcomings of PLA and TPS. Also pictures of produced films should be provided, as this is a clear indicator of “their suitability for applications in the packaging industry”.

Authors: According to the reviewer's comment, pictures of films produced were presented - Figure 7 was added, and the numbers of other Figures were changed

Remark # 3: “-Figures 5 and 6: how are error bars computed? Are they computed from spatial variations within a single test strip? This also connect to a previous comment on bubble stability: since error bars are significantly smaller for LDPE, this means that the corresponding bubble and thus process is much more stable. This needs to be stated."

Authors: In the 'Conclusions' chapter, an additional paragraph has been introduced to comment on the differences in the size of the error bars between PE and biodegradable materials with an indication of the cause and consequences that may result from this.

Remark # 4: “Though the new presentation of MFR results is far more instructive, the caption to Figure 4 is wrong, as there is no screw speed effect. The discussion of MFR results does not connect to the discussion of extrusion rate and flowability. This is critically missing, as my previous comment on LDPE flowability is not addressed in the revised paper. In the abstract, it should be made clear that results for films were obtained under specific screw speed and temperatures conditions. This is because conclusions could be drastically different for other processing conditions.”

Authors: The figure showing the MFR has been corrected - the speed description has been deleted.

A description has been introduced into the abstract, clearly indicating that the films were extruded at individually set temperatures and had fixed geometrical features.

Remark # 5: “-line 71 “extremely good processability”: this statement for PLA is at odd with the numerous literature on PLA modification for improving its film blowing ability, see for instance Kharrat, Chaari et al, J. Renewable Materials 2019. So these lines should be revised, separating PLA thermal and mechanical properties, from its processability in film blowing. In other words, it should be made clear that blowing PLA into a film is impossible, unless properly modified using different strategies.”

Authors: In fact, an acronym is used which is not very clear to the reader, bearing in mind that it is possible to achieve very good processability through various types of modifications, as few commercially available granules do not have them. The sentence has been rewritten so that there is no doubt as to its correctness.

Remark # 6: “-Figure 8, vertical axis: units are wrong for a speed.”

Authors: The units have been corrected.

Remark # 7: “-line 688, “flowability”: how does flowability relate with MFR of LDPE at processing temperatures? The comment on LDPE flowability seems at odd with MFR values which are the largest.”

Authors: A potential reason for discrepancies in flow velocity and melt flow rates at processing temperatures may be the effect of shear resulting from the rotational motion of the screw, especially since the speeds used were quite high. The MFR charts show a significant increase in the melt flow rate of biodegradable materials with increasing temperature, in particular in the case of PLA, which at 190 ° C had a very good rate. The shear effect can cause an autothermal effect, whereby the local material mass temperature can be much higher than the current barrel wall temperature, which can potentially result in even a drastic increase in melt flow rate. Measurement of the rate allows you to get an idea of ​​how the processability of materials is shaped, however, due to the static nature of the measurement, it does not fully reflect what is happening in the plasticizing system. The effect of processing on the physical properties of materials is confirmed by the analysis of DSC results of the first heating cycle for granules and films. The processing process practically did not affect the thermal properties of PE, the values ​​of Tg, Tm, ΔH and X for the granulate and the obtained films are comparable. Clear differences can be seen for materials based on potato and corn starch. The TPS-C film has twice higher ΔH and X in relation to its granulate, and the TPS-P film three times. This proves the significant impact of processing on the properties of these materials.

The relevant paragraph has been added to the text.

Best regards,

Authors